# STATE-REGULARIZED RECURRENT NETWORKS

## ABSTRACT

Recurrent networks are a widely used class of neural architectures. They have, however, two shortcomings. First, it is difficult to understand what exactly they learn. Second, they tend to work poorly on sequences requiring long-term memorization, despite having this capacity in principle. We aim to address both shortcomings with a class of recurrent networks that use a stochastic state transition mechanism between cell applications. This mechanism, which we term state-regularization, makes RNNs transition between a finite set of learnable states. We show that state-regularization (a) simplifies the extraction of finite state automata modeling an RNN's state transition dynamics, and (b) forces RNNs to operate more like automata with external memory and less like finite state machines.

## 1 INTRODUCTION

Recurrent neural networks (RNNs) have found their way into numerous applications. Still, RNNs have two shortcomings. First, it is difficult to understand what concretely RNNs learn. Some applications require a close inspection of learned models before deployment and RNNs are more difficult to interpret than rule-based systems. There are a number of approaches for extracting finite state automata (DFAs) from trained RNNs (Giles et al., 1991; Wang et al., 2018b; Weiss et al., 2018b) as a means to analyze their behavior. These methods apply extraction algorithms after training and it remains challenging to determine whether the extracted DFA faithfully models the RNN's state transition behavior. Most extraction methods are rather complex, depend crucially on hyperparameter choices, and tend to be computationally costly. Second, RNNs tend to work poorly on input sequences requiring long-term memorization, despite having this ability in principle. Indeed, there is a growing body of work providing evidence, both empirically (Daniluk et al., 2017a; Bai et al., 2018) and theoretically (Miller & Hardt, 2018), that recurrent networks offer no benefit on longer sequences, at least under certain conditions. Intuitively, RNNs tend to operate more like DFAs with a large number of states, attempting to memorize all the information about the input sequence solely with their hidden states, and less like automata with external memory.

We propose state-regularized RNNs as a possible step towards addressing both of the aforementioned problems. State-regularized RNNs (SR-RNNs) are a class of recurrent networks that utilize a stochastic state transition mechanism between cell applications. The stochastic mechanism models a probabilistic state dynamics that lets the SR-RNN transition between a finite number of learnable states. The parameters of the stochastic mechanism are trained jointly with the parameters of the base RNN. SR-RNNs have several advantages over standard RNNs. Instead of having to apply post-training DFA extraction, it is possible to determine their (probabilistic and deterministic) state transition behavior more directly. We propose a method that extracts DFAs truly representing the state transition behavior of the underlying RNNs. We hypothesize that the frequently-observed poor extrapolation behavior of RNNs is caused by memorization with hidden states. It is known that RNNs – even those with cell states or external memory – tend to memorize mainly with their hidden states and in an unstructured manner (Strobelt et al., 2016; Hao et al., 2018). We show that the state-regularization mechanism shifts representational power to memory components such as the cell state, resulting in improved extrapolation performance.

We support our hypotheses through experiments both on synthetic and real-world datasets. We explore the improvement of the extrapolation capabilities of SR-RNNs and closely investigate their memorization behavior. For state-regularized LSTMs, for instance, we observe that memorization can be shifted entirely from the hidden state to the cell state. For text and visual data, state-regularization provides more intuitive interpretations of the RNNs' behavior.

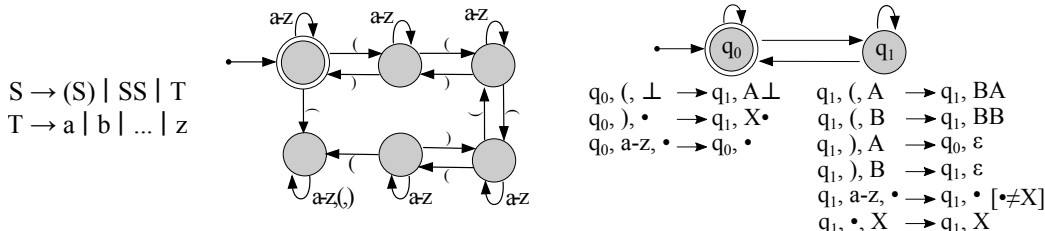

Figure 1: (Left) The context-free grammar for the language balanced parentheses (BP). (Center) A DFA that recognizes BP up to depth 4. (Right) A deterministic pushdown automaton (DPDA) that recognizes BP for all depths. The symbol • is a wildcard and stands for all possible tokens. The DPDA extrapolates to all sequences of BP, the DFA recognizes only those up to nesting depth 4.

## 2  BACKGROUND

We provide some background on deterministic finite automata (DFAs) and deterministic pushdown automata (DPDAs) for two reasons. First, one contribution of our work is a method for extracting DFAs from RNNs. Second, the state regularization we propose is intended to make RNNs behave more like DPDAs and less like DFAs by limiting their ability to memorize with hidden states.

A DFA is a state machine that accepts or rejects sequences of tokens and produces one unique computation path for each input. Let $\Sigma^*$ be the language over the alphabet $\Sigma$ and let $\epsilon$ be the empty sequence. A DFA over an alphabet (set of tokens) $\Sigma$ is a 5-tuple $(\mathcal{Q}, \Sigma, \delta, q_0, F)$ consisting of finite set of states $\mathcal{Q}$; a finite set of input tokens $\Sigma$ called the input alphabet; a transition functions $\delta : \mathcal{Q} \times \Sigma \to \mathcal{Q}$; a start state $q_0$; and a set of accept states $F \subseteq \mathcal{Q}$. A sequence $w$ is accepted by the DFA if the application of the transition function, starting with $q_0$, leads to an accepting state. Figure 1(center) depicts a DFA for the language of balanced parentheses (BP) up to depth 4. A language is regular if and only if it is accepted by a DFA.

A pushdown automata (PDA) is defined as a 7-tuple $(\mathcal{Q}, \Sigma, \Gamma, \delta, q_0, \perp, F)$ consisting of a finite set of states $\mathcal{Q}$; a finite set of input tokens $\Sigma$ called the input alphabet; a finite set of tokens $\Gamma$ called the stack alphabet; a transition function $\delta \subseteq \mathcal{Q} \times (\Sigma \cup \epsilon) \times \Gamma \to \mathcal{Q} \times \Gamma^*$; a start state $q_0$; the initial stack symbol $\perp$; and a set of accepting states $F \subseteq \mathcal{Q}$. Computations of the PDA are applications of the transition relations. The computation starts in $q_0$ with the initial stack symbol $\perp$ on the stack and sequence $w$ as input. The pushdown automaton accepts $w$ if after reading $w$ the automaton reaches an accepting state. Figure 1(right) depicts a deterministic PDA for the language BP.

## 3  STATE-REGULARIZED RECURRENT NEURAL NETWORKS

The standard recurrence of an RNN is $\mathbf{h}_t = f(\mathbf{h}_{t-1}, \mathbf{c}_{t-1}, \mathbf{x}_t)$ where $\mathbf{h}_t$ is the hidden state vector at time $t$, $\mathbf{c}_t$ is the cell state at time $t$, and $\mathbf{x}_t$ is the input symbol at time $t$. We refer to RNNs whose unrolled cells are only connected through the hidden output states $\mathbf{h}$ and no additional vectors such as the cell state, as *memory-less* RNNs. For instance, the family of GRUs (Chung et al., 2014) does not have cell states and, therefore, is memory-less. LSTMs (Hochreiter & Schmidhuber, 1997), on the other hand, are not memory-less due to their cell state.

A cell of a state-regularized RNN (SR-RNN) consist of two components. The first component, which we refer to as the *recurrent component*, applies the function of a standard RNN cell

$$\mathbf{u}_t = f(\mathbf{h}_{t-1}, \mathbf{c}_{t-1}, \mathbf{x}_t).$$

For the sake of completeness, we include the cell state $\mathbf{c}$ here, which is absent in memory-less RNNs.

We propose a second component which we refer to as *stochastic component*. The stochastic component is responsible for modeling the probabilistic state transitions mechanism that lets the RNN transition implicitly between a finite number of states. Let $d$ be the size of the hidden state vectors of the recurrent cells. Moreover, let $\Delta^D := \{\boldsymbol{\lambda} \in \mathbb{R}_+^D : \|\boldsymbol{\lambda}\| = 1\}$ be the $(D-1)$ probability simplex. The stochastic component maintains $k$ learnable centroids $\mathbf{s}_1, ..., \mathbf{s}_k$ of size $d$ which we often write as the column vectors of a matrix $\mathbf{S} \in \mathbb{R}^{d \times k}$. The weights of these centroids are global parameters

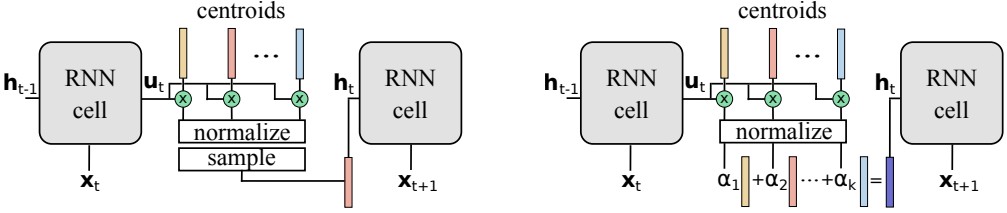

Figure 2: Two possible instances of an SR-RNN corresponding to equations 2&4 and 2&5.

shared among all cells. The stochastic component computes, at each time step $t$, a discrete probability distribution from the output $\mathbf{u}_t$ of the recurrent component and the centroids of the stochastic component

$$\boldsymbol{\alpha} = g(\mathbf{S}, \mathbf{u}_t) \text{ with } \boldsymbol{\alpha} \in \Delta^k. \tag{1}$$

Crucially, instances of $g$ should be differentiable to facilitate end-to-end training. Typical instances of the function $g$ are based on the dot-product or the Euclidean distance, normalized into a probability distribution

$$\alpha_i = \frac{\exp\left((\mathbf{u}_t \cdot \mathbf{s}_i)/\tau\right)}{\sum_{i=1}^{k} \exp\left((\mathbf{u}_t \cdot \mathbf{s}_i)/\tau\right)} \tag{2} \qquad \alpha_i = \frac{\exp\left(-\parallel \mathbf{u}_t - \mathbf{s}_i \parallel /\tau\right)}{\sum_{i=1}^{k} \exp\left(-\parallel \mathbf{u}_t - \mathbf{s}_i \parallel /\tau\right)} \tag{3}$$

Here, $\tau$ is a temperature parameter that can be used to anneal the probabilistic state transition behavior. The lower $\tau$ the more $\boldsymbol{\alpha}$ resembles the one-hot encoding of a centroid. The higher $\tau$ the more uniform is $\boldsymbol{\alpha}$. Equation 2 is reminiscent of the equations of attentive mechanisms (Bahdanau et al., 2015; Vaswani et al., 2017). Instead of attending to the hidden states, however, SR-RNNs attend to the $k$ centroids to compute transition probabilities. Each $\alpha_i$ is the probability of the RNN to transition to centroid (state) $i$ given the vector $\mathbf{u}_t$ for which we write $p_{\mathbf{u}_t}(i) = \alpha_i$.

The state transition dynamics of an SR-RNN is that of a probabilistic finite state machine. At each time step, when being in state $\mathbf{h}_{t-1}$ and reading input symbol $\mathbf{x}_t$, the probability for transitioning to state $\mathbf{s}_i$ is $\alpha_i$. Hence, in its second phase the stochastic component computes the hidden state $\mathbf{h}_t$ at time step $t$ from the distribution $\boldsymbol{\alpha}$ and the matrix $\mathbf{S}$ with a (possibly stochastic) mapping $h : \Delta^k \times \mathbb{R}^{d \times k} \to \mathbb{R}^d$. Hence, $\mathbf{h}_t = h(\boldsymbol{\alpha}, \mathbf{S})$. A simple instance of the mapping $h$ is to

$$\text{sample } j \sim p_{\mathbf{u}_t} \text{ and set } \mathbf{h}_t = \mathbf{s}_j. \tag{4}$$

This renders the SR-RNN not end-to-end differentiable, however, and one has to use EM or reinforcement learning strategies which are often less stable and less efficient. A possible alternative is to set the hidden state $\mathbf{h}_t$ to be the probabilistic mixture of the centroids

$$\mathbf{h}_t = \sum_{i=1}^{k} \alpha_i \mathbf{s}_i. \tag{5}$$

Every internal state $\mathbf{h}$ of the SR-RNN, therefore, is computed as a weighted sum $\mathbf{h} = \alpha_1 \mathbf{s}_1 + ... + \alpha_k \mathbf{s}_k$ of the centroids $\mathbf{s}_1, ..., \mathbf{s}_k$ with $\boldsymbol{\alpha} \in \Delta^k$. Here, $h$ is a smoothed variant of the function that computes a hard assignment to one of the centroids. One can show that for $\tau \to 0$ the state dynamics based on equations (4) and (5) are identical and correspond to those of a DFA. Figure 2 depicts two variants of the proposed SR-RNNs.

Additional instances of $h$ are conceivable. For instance, one could, for every input sequence and the given current parameters of the SR-RNN, compute the most probable state sequence and then backpropagate based on a structured loss. Since finding these most probable sequences is possible with Viterbi type algorithms, one can apply a form of differentiable dynamic programming (Mensch & Blondel, 2018). The probabilistic state transitions of SR-RNNs open up new possibilities for applying more complex differentiable functions. We leave these considerations to future work. The probabilistic state transition mechanism is also applicable when RNNs have more than one hidden layer. In RNNs with $l > 1$ hidden layers, every such layer can maintain its own centroids and stochastic component. In this case, a global state of the SR-RNN is an $l$-tuple, with the $l$th argument of the tuple corresponding to the centroids of the $l$th layer.

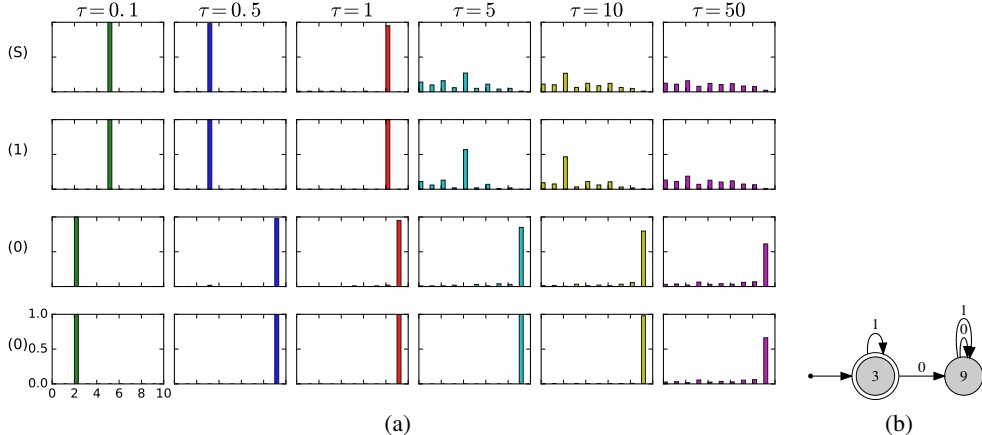

(a)                                                                (b)

Figure 3: (Left) State transition probabilities for the SR-GRU learned from the data for the Tomita 1 grammar, for temperatures $\tau$ and input 100. S is the start token. States are listed on x-axis, probabilities on y-axis. Up to temperature $\tau = 1$ the behavior of the trained SR-GRUs is almost identical to that of a DFA. Despite the availability of 10 centroids, the trained SR-GRUs use the minimal set of states for $\tau \leq 1$. (Right) The extracted DFA for Tomita grammar 1 and temperature $\tau = 0.5$.

Even though we have augmented the original RNN with additional learnable parameter vectors, we are actually constraining the SR-RNN to output hidden state vectors that are similar to the centroids. For lower temperatures and smaller values for $k$, the ability of the SR-RNN to memorize with its hidden states is increasingly impoverished. We argue that this behavior is beneficial for three reasons. First, it makes the extraction of interpretable DFAs from memory-less SR-RNNs straightforward. Instead of applying post-training DFA extraction as in previous work, we extract the true underlying DFA directly from the SR-RNN. Second, we hypothesize that overfitting in the context of RNNs is often caused by memorization via hidden states. Indeed, we show that regularizing the state space pushes representational power to memory components such as the cell state of an LSTM, resulting in improved extrapolation behavior. Third, the values of hidden states tend to increase in magnitude with the length of the input sequence, a behavior that has been termed *drifting* (Zeng et al., 1993). The proposed state regularization stabilizes the hidden states for longer sequences.

First, let us explore some of the theoretical properties of the proposed mechanism. We show that the addition of the stochastic component, when capturing the complete information flow between cells as, for instance, in the case of GRUs, makes the resulting RNN's state transition behavior identical to that of a probabilistic finite state machine. The proofs of the theorems are part of the appendix.

**Theorem 3.1.** *The state transition behavior of a memory-less* SR-RNN *using equation 4 is identical to that of a probabilistic finite automaton.*

We can show that the lower the temperature the more memory-less RNNs operate like DFAs.

**Theorem 3.2.** *For $\tau \to 0$ the state transition behavior of a memory-less* SR-RNN *(using equations 4 or 5) is equivalent to that of a deterministic finite automaton.*

## 3.1 LEARNING DFAS WITH STATE-REGULARIZED RNNS

Extracting DFAs from RNNs is motivated by applications where a thorough understanding of learned neural models is required before deployment. SR-RNNs maintain a set of learnable states and compute and explicitly follow state transition probabilities. It is possible, therefore, to extract finite-state transition functions that truly model the underlying state dynamics of the SR-RNN. The centroids do not have to be extracted from a clustering of a number of observed hidden states but can be read off of the trained model. This renders the extraction also more efficient. We adapt previous work (Schellhammer et al., 1998; Wang et al., 2018b) to construct the transition function of a SR-RNN. We begin with the start token of an input sequence, compute the transition probabilities $\alpha$, and move the SR-RNN to the highest probability state. We continue this process until we have seen the last input token. By doing this, we get a count of transitions from every state $\mathbf{s}_i$ and input

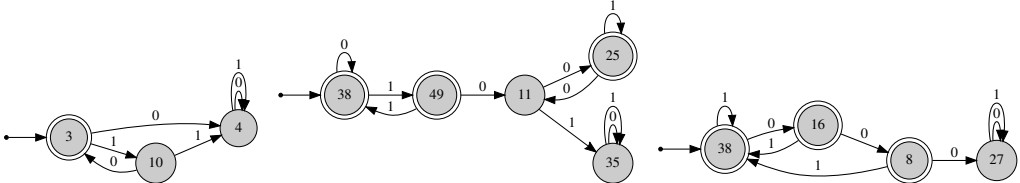

Figure 4: DFAs corresponding to the Tomita grammars 2-4. The numbers on the states correspond directly to the centroid numbers of the learned SR-GRU.

token $a \in \Sigma$ to the following states (including selfloops). After obtaining the transition counts, we keep only the most frequent transitions and discard all other transitions. Due to space constraints, the pseudo-code of the extraction algorithm is listed in the appendix. As a corollary of Theorem 3.2 we have that, for $\tau \to 0$, the extracted transition function is identical to the transition function of the DFA learned by the SR-RNN. Figure 3 shows that for a wide range of temperatures (including the standard softmax temperature $\tau = 1$) the transition behavior of a SR-GRU is identical to that of a DFA, a behavior we can show to be common when SR-RNNs are trained on regular languages.

### 3.2 LEARNING NONREGULAR LANGUAGES WITH STATE-REGULARIZED LSTMS

For more complex languages such as context-free languages, RNNs that behave like DFAs generalize poorly to longer sequences. The DPDA shown in Figure 1, for instance, correctly recognizes the language of BP while the DFA only recognizes it up to nesting depth 4. We want to encourage RNNs with memory to behavor more like DPDAs and less like DFAs. The transition function $\delta$ of a DPDA takes (a) the current state, (b) the current top stack symbol, and (c) the current input symbol and maps these inputs to (1) a new state and (2) a replacement of the top stack symbol (see section 2). Hence, to allow an SR-RNN such as the SR-LSTM to operate in a manner similar to a DPDA we need to give the RNNs access to these three inputs when deciding what to forget from and what to add to memory. Precisely this is accomplished for LSTMs with peephole connections (Gers & Schmidhuber, 2000). The following additions to the functions defining the forget, input, and output gates include the cell state into the LSTM's memory update decisions

$$\text{forget gate:} \quad \mathbf{f}_t = \sigma\big(\mathbf{W}^f\mathbf{x}_t + \mathbf{R}^f\mathbf{h}_{t-1} + \mathbf{p}^f \odot \mathbf{c}_{t-1} + \mathbf{b}^f\big) \tag{6}$$

$$\text{input gate:} \quad \mathbf{i}_t = \sigma\big(\mathbf{W}^i\mathbf{x}_t + \mathbf{R}^i\mathbf{h}_{t-1} + \mathbf{p}^i \odot \mathbf{c}_{t-1} + \mathbf{b}^i\big) \tag{7}$$

$$\text{output gate:} \quad \mathbf{o}_t = \sigma\big(\mathbf{W}^o\mathbf{x}_t + \mathbf{R}^o\mathbf{h}_{t-1} + \mathbf{p}^o \odot \mathbf{c}_t + \mathbf{b}^o\big). \tag{8}$$

Here, $\mathbf{h}_{t-1}$ is the output of the previous cell's stochastic component; $\mathbf{W}$s and $\mathbf{R}$s are the matrices of the original LSTM; the $\mathbf{p}$s are the parameters of the peephole connections; and $\odot$ is the elementwise multiplication. We show empirically that the resulting SR-LSTM-P operates like a DPDA, incorporating the current cell state when making decisions about changes to the next cell state.

### 3.3 PRACTICAL CONSIDERATIONS

Implementing SR-RNNs only requires extending existing RNN cells with a stochastic component. We have found the use of start and end tokens to be beneficial. The start token is used to transition the SR-RNN to a centroid representing the start state which then does not have to be fixed a priori. The end token is used to perform one more cell application but without applying the stochastic component before a classification layer. The end token lets the SR-RNN consider both the cell state and the hidden state to make the accept/reject decision. We find that a temperature of $\tau = 1$ (standard softmax) and an initialization of the centroids with values sampled uniformly from $[-0.5, 0.5]$ work well across different datasets.

## 4 EXPERIMENTS

We conduct three types of experiments to investigate our hypotheses. First, we apply a simple algorithm for extracting DFAs and assess to what extent the true DFAs can be recovered from input data. Second, we compare the behavior of LSTMs and state-regularized LSTM on nonregular

| Dataset | Large Dataset | | | Small Dataset | | |
|---|---|---|---|---|---|---|
| Models | LSTM | SR-LSTM | SR-LSTM-P | LSTM | SR-LSTM | SR-LSTM-P |
| $d \in [1, 10], l \leq 100$ | 0.005 | 0.038 | **0.000** | 0.068 | 0.037 | **0.017** |
| $d \in [10, 20], l \leq 100$ | 0.334 | 0.255 | **0.001** | 0.472 | 0.347 | **0.189** |
| $d \in [10, 20], l \leq 200$ | 0.341 | 0.313 | **0.003** | 0.479 | 0.352 | **0.196** |
| $d = 5, l \leq 200$ | 0.002 | 0.044 | **0.000** | 0.042 | 0.028 | **0.015** |
| $d = 10, l \leq 200$ | 0.207 | 0.227 | **0.004** | 0.409 | 0.279 | **0.138** |
| $d = 20, l \leq 1000$ | 0.543 | 0.540 | **0.020** | 0.519 | 0.508 | **0.380** |

Table 1: Error rates for the balanced parentheses (BP) test sets ($d$=depth, $l$=length, $k = 5$).

languages such as the languages of balanced parentheses and palindromes. Third, we investigate the performance of state-regularized LSTMs on non-synthetic datasets.

Unless otherwise indicated we always (a) use single-layer RNNs, (b) learn an embedding for input tokens before feeding it to the RNNs, (c) apply RMSPROP with a learning rate of $0.01$ and momentum of $0.9$; (d) do not use dropout or batch normalization of any kind; and (e) use state-regularized RNNs based on equations 2&5 with a temperature of $\tau = 1$ (standard softmax).

### 4.1 REGULAR LANGUAGES AND DFA EXTRACTION

We evaluate the DFA extraction algorithm for SR-RNNs on RNNs trained on the Tomita grammars (Tomita, 1982) which have been used as benchmarks in previous work (Wang et al., 2018b; Weiss et al., 2018b). We use available code (Weiss et al., 2018b) to generate training and test data for the regular languages. We first trained a single-layer GRU with $100$ units on the data. We use GRUs since they are memory-less and, hence, Theorem 3.2 applies. Whenever the GRU converged within 1 hour to a training accuracy of $100\%$, we also trained a SR-GRU based on equations 2&5 with $k = 50$ and $\tau = 1$. This was the case for the grammars 1-4 and 7. The difference in time to convergence between the vanilla GRU and the SR-GRU was negligible. We applied the transition function extraction outlined in section 3.1. In all cases, we could recover the minimal and correct DFA corresponding to the grammars. Figure 4 depicts the DFAs for grammars 2-4 extracted by our approach. Remarkably, even though we provide more centroids (possible states; here $k = 50$) the SR-GRU only utilizes the required minimal number of states for each of the grammars. Figure 3 visualizes the transition probabilities for different temperatures and $k = 10$ for grammar 1. The numbers on the states correspond directly to the centroid numbers of the learned SR-GRU. One can observe that the probabilities are spiky, causing the SR-GRU to behave like a DFA for $\tau \leq 1$.

### 4.2 NONREGULAR LANGUAGES

We conducted experiments on nonregular languages where external memorization is required. We wanted to investigate whether our hypothesis that SR-LSTM behave more like DPDAs and, therefore, extrapolate to longer sequences, is correct. To that end, we used the context-free language "balanced parentheses" (BP; see Figure 1(left)) over the alphabet $\Sigma = \{a, ..., z, (, )\}$, used in previous work (Weiss et al., 2018b). We created two datasets for BP. A large one with 22,286 training sequences (positive: 13,025; negative: 9,261) and 6,704 validation sequences (positive: 3,582; negative: 3,122). The small dataset consists of 1,008 training sequences (positive: 601; negative: 407), and 268 validation sequences (positive: 142; negative: 126). The training sequences have nesting depths $d \in [1, 5]$ and the validation sequences $d \in [6, 10]$. We trained the LSTM and the SR-RNNs using curriculum learning as in previous work (Zaremba & Sutskever, 2014; Weiss et al., 2018b) and using the validation error as stopping criterion. We then applied the trained models to unseen sequences. Table 1 lists the results on 1,000 test sequences each with the respective depths and lengths. The results show that both SR-LSTM and SR-LSTM-Ps extrapolate better to longer sequences and sequences with deeper nesting. Moreover, the SR-LSTM-P performs almost perfectly on the large data indicating that peephole connections are indeed beneficial.

To explore the effect of the hyperparameter $k$, that is, the number of centroids of the SR-RNNs, we ran experiments on the small BP dataset varying $k$ and keeping everything else the same. Table 2 lists the error rates and Figure 5 the error curves on the validation data for the SR-LSTM-P and different values of $k$. While two centroids ($k = 2$) result in the best error rates for most sequence

| Number of centroids | $k = 2$ | $k = 5$ | $k = 10$ | $k = 50$ |
|---|---|---|---|---|
| $d \in [1, 10], l \leq 100$ | 0.019 | **0.017** | 0.021 | 0.034 |
| $d \in [10, 20], l \leq 100$ | **0.096** | 0.189 | 0.205 | 0.192 |
| $d \in [10, 20], l \leq 200$ | **0.097** | 0.196 | 0.213 | 0.191 |
| $d = 5, l \leq 200$ | 0.014 | 0.015 | **0.012** | 0.047 |
| $d = 10, l \leq 200$ | **0.038** | 0.138 | 0.154 | 0.128 |
| $d = 20, l \leq 1000$ | 0.399 | **0.380** | 0.432 | 0.410 |

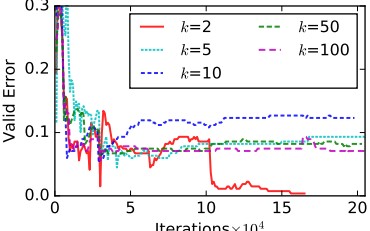

Table 2: Error rates of the SR-LSTM-P on the small BP test data for various numbers of centroids ($d$=depth, $l$=length).

Figure 5: SR-LSTM-P error curves on the small BP validation data.

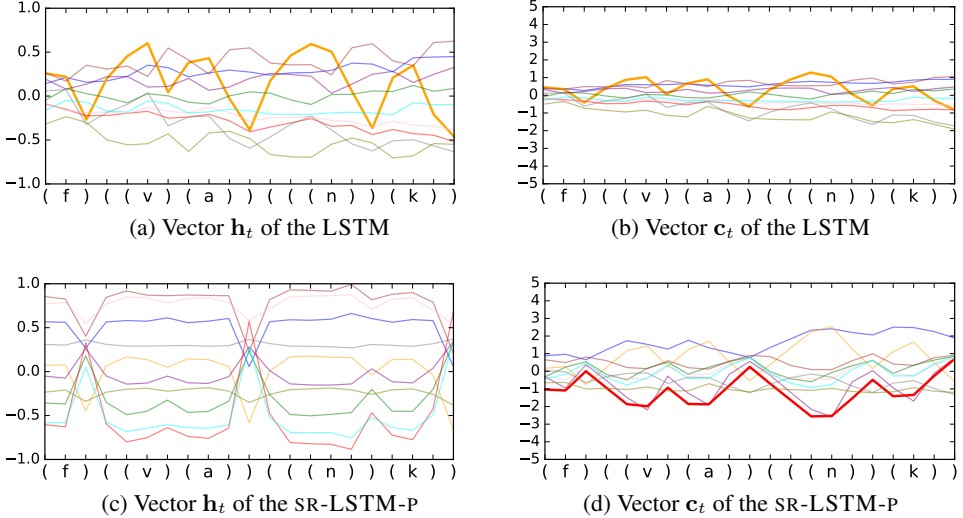

(a) Vector $\mathbf{h}_t$ of the LSTM

(b) Vector $\mathbf{c}_t$ of the LSTM

(c) Vector $\mathbf{h}_t$ of the SR-LSTM-P

(d) Vector $\mathbf{c}_t$ of the SR-LSTM-P

Figure 6: Visualization of hidden state $\mathbf{h}_t$ and cell state $\mathbf{c}_t$ of the LSTM and the SR-LSTM-P for a specific input sequence from BP. The LSTM memorizes the number of open parentheses both in the hidden and to a lesser extent in the cell state (bold yellow lines). The memorization is not accomplished with saturated gate outputs and a drift is observable for both vectors. The SR-LSTM-P maintains two distinct hidden states (accept and reject) and does not visibly memorize counts through its hidden states. The cell state is used to cleanly memorize the number of open parentheses (bold red line) with saturated gate outputs ($\pm1$). A state vector drift is not observable.

types, the differences are not very pronounced. This indicates that the SR-LSTM-P is robust to changes in the hyperparameter $k$. A close inspection of the transition probabilities reveals that the SR-LSTM-P mostly utilizes two states, independent of the value of $k$. These two states are used as accept and reject states. These results show that SR-RNNs generalize tend to utilize a minimal set states similar to DPDAs.

A major hypothesis of ours is that the state-regularization encourages RNNs to operate more like DPDAs. To explore this hypothesis, we trained an SR-LSTM with 10 units on the BP data and visualized both the hidden state $\mathbf{h}_t$ and the cell state $\mathbf{c}_t$ for various input sequences. Similar state visualizations have been used in previous work (Strobelt et al., 2016; Weiss et al., 2018a). Figure 6 plots the hidden and cell states for a specific input, where each color corresponds to a dimension in the respective state vectors. As hypothesized, the LSTM relies primarily on its hidden states for memorization. The SR-LSTM-P, on the other hand, does not use its hidden states for memorization. Instead it utilizes two main states (accept and reject) and memorizes the nesting depth cleanly in the cell state. The visualization also shows a drifting behavior for the LSTM, in line with observations made for first-generation RNNs (Zeng et al., 1993). Drifting is not observable for the SR-LSTM-P.

We also performed experiments for the nonregular language $ww^{-1}$ (Palindromes) over the alphabet $\Sigma = \{a, ..., z\}$. We follow the same experiment setup as for BP and include the details in the appendix. The results of Table 3 add evidence for the improved generalization behavior of state-regularized LSTMs. We will make available all datasets used in the experiments.

| max length | 100 | 200 | 500 |
|---|---|---|---|
| LSTM | 31.2 | 42.0 | 47.7 |
| SR-LSTM | 28.0 | 36.0 | 44.6 |
| SR-LSTM-P | **10.5** | **16.7** | **29.8** |

| model length | LSTM | | SR-LSTM | | SR-LSTM-P | |
|---|---|---|---|---|---|---|
| | 100 | 200 | 100 | 200 | 100 | 200 |
| train error | 29.23 | 32.77 | 27.17 | 31.72 | **23.30** | **24.67** |
| test error | 29.96 | 33.19 | 28.82 | 32.88 | **26.04** | **26.21** |
| time | 400.29s/epoch | | 429.15s/epoch | | 466.10s/epoch | |

Table 3: Error rates in % on sequences of varying lengths from the Palindrome test set.

Table 4: Error rates (on training and test splits of the IMDB data) in % and averaged training time in seconds when training only on truncated sequences of length 10.

| Methods | Error |
|---|---|
| **use additional unlabeled data** | |
| Full+unlabelled+BoW (Maas et al.) | 11.1 |
| LM-LSTM+unlabelled (Dai & Le) | 7.6 |
| SA-LSTM+unlabelled (Dai & Le) | 7.2 |
| **do not use additional unlabeled data** | |
| seq2-bown-CNN (Johnson & Zhang) | 14.7 |
| WRRBM+BoW(bnc) (Dahl et al.) | 10.8 |
| JumpLSTM (Yu et al.) | 10.6 |
| LSTM (max pool) | 10.1 |
| LSTM-P (max pool) | 10.3 |
| SR-LSTM (max pool, $k = 10$) | 9.4 |
| SR-LSTM-P (max pool, $k = 50$) | 9.8 |
| SR-LSTM-P (max pool, $k = 10$) | **9.2** |
| SR-LSTM-P (2 layer, last, $c^1$, $c^2$, $k$=10) | 9.4 |

| Methods | Accuracy | |
|---|---|---|
| | Normal | Perm. |
| IRNN (Le et al.) | 97.0 | 82.0 |
| URNN (Arjovsky et al.) | 95.1 | 88.0 |
| sTANH-RNN [Zhang *et al.*, 2016] | 98.1 | 94.0 |
| RWA (Ostmeyer & Cowell) | 98.1 | 93.5 |
| Skip LSTM (Campos et al.) | 97.3 | — |
| BN-LSTM (Cooijmans et al.) | 99.0 | **95.4** |
| Zoneout (Krueger et al.) | — | 93.1 |
| Dilated GRU (Chang et al.) | **99.2** | 94.6 |
| LSTM | 97.7 | 91.2 |
| LSTM-P | 98.5 | 91.9 |
| SR-LSTM ($k = 100$) | 98.6 | 91.6 |
| SR-LSTM-P ($k = 100$) | **99.2** | 92.1 |
| SR-LSTM-P ($k = 50$) | 98.6 | 91.5 |
| SR-LSTM-P ($k = 10$) | 98.1 | 91.0 |

Table 5: Error rates in % of some SR-LSTM-Ps and state of the art methods for IMDB.

Table 7: Accuracy in % of some SR-LSTM-Ps and state of the art methods for sequential MNIST.

## 4.3 SENTIMENT ANALYSIS AND PIXEL-BY-PIXEL MNIST

We evaluated state-regularized LSTMs on the IMDB review dataset (Maas et al., 2011). It consists of 100k movie reviews (25k training, 25k test, and 50k unlabeled). We used only the labeled training and test reviews. Each review is labeled as *positive* or *negative*. To investigate the models' extrapolation capabilities, we trained all models on truncated sequences of up to length 10 but tested on longer sequences (length 100 and 200). The results listed in Table 4 show the improvement in extrapolation performance of the SR-LSTM-P and, to a lesser extent, the SR-LSTM. The table also lists the training time per epoch, indicating that the overhead compared to the standard LSTM is modest. Table 5 lists the results when training without truncating sequences. The SR-LSTM-P is competitive with state of the art methods that also do not use the unlabeled data.

We explored the impact of state-regularization on the pixel-by-pixel MNIST digits problem (Le et al., 2015). Here, the 784 pixels of MNIST images are fed to RNNs one by one for classification. This requires the modeling of long-term dependencies. Table 7 shows the results on the normal and permuted versions of pixel-by-pixel MNIST. The classification function has the final hidden and cell state as input. Our (state-regularized) LSTMs do not use dropout, batch normalization, sophisticated weight-initialization, and are based on a simple single-layer LSTM. While the SR-LSTM-P cannot reach state of the art accuracy, we observe an improvement compared to the LSTM.

State regularization provides new ways to interpret the working of recurrent networks[1]. Since SR-RNNs have a finite set of states, we can use the observed transition probabilities to visualize their behavior. For instance, to generate prototypes for the SR-RNNs we can select, for each state $i$, the input tokens that have the highest average probability leading to state $i$. For the IMDB reviews, these are the top probability words leading to each of the states. For pixel-by-pixel MNIST, these are the top probabilities of each of the pixels leading to the states. Table 8 lists, for each state (centroid), the word with the top transition probabilities leading to this state. Figure 7 shows the prototypes associated with 4 states for the MNIST digits "3" and "7". In contrast to previous methods (Berkes & Wiskott, 2006; Nguyen et al., 2016), the prototypes are directly generated with the centroids of the trained SR-RNNs and the input token embeddings and hidden states do not have to be similar.

---

[1] We use the definitions of explainability and interpretability of previous work (Montavon et al., 2017).

| cent. | words with top-4 highest transition probabilities |
|---|---|
| 1 | but (0.97) hadn (0.905) college (0.87) even (0.853) |
| 2 | not (0.997) or (0.997) italian (0.995) never (0.993) |
| 3 | loved (0.998) definitely (0.996) 8 (0.993) realistic (0.992) |
| 4 | no (1.0) worst (1.0) terrible (1.0) poorly (1.0) |

Table 8: The words with the top-4 highest transition probabilities for four centroids. This visualizes the SR-LSTM-P's behavior and associates centroids with prototypical words.

Figure 7: Four prototypes generated from SR-LSTM-P ($k = 10$) centroids (0,4,5,6) for MNIST.

## 5 RELATED WORK

RNNs are powerful learning machines. Siegelmann and Sontag (Siegelmann & Sontag, 1992; 1994; Siegelmann, 2012), for instance, proved that a variant of Elman-RNNs (Elman, 1990) can simulate a Turing machine. Recent work considers the more practical situation where RNNs have finite precision and linear computation time in their input length (Weiss et al., 2018a).

Extracting DFAs from RNNs goes back to work on first-generation RNNs in the 1990s(Giles et al., 1991; Zeng et al., 1993). These methods perform a clustering of hidden states after the RNNs are trained (Wang et al., 2018a; Frasconi & Bengio, 1994; Giles et al., 1991). Recent work introduced more sophisticated learning approaches to extract DFAs from LSTMs and GRUs (Weiss et al., 2018b). The latter methods tend to be more successful in finding DFAs behaving similar to the RNN. In contrast to all existing methods, SR-RNN learn an explicit set of states which facilitates the extraction of DFAs from memory-less SR-RNNs exactly modeling their state transition dynamics. A different line of work attempt to learn a more interpretable rule-based classifier from RNNs (Murdoch & Szlam, 2017).

There is a large body of work on regularization techniques for RNNs. Most of these adapt regularization approaches developed for feed-forward networks to the recurrent setting. Representative instances are dropout regularization (Zaremba et al., 2014), variational dropout (Gal & Ghahramani, 2016), weight-dropped LSTMs (Merity et al., 2018), and noise injection (Dieng et al., 2018). Two approaches that can improve convergence and generalization capabilities are batch normalization (Cooijmans et al., 2017) and weight initialization strategies (Le et al., 2015) for RNNs. The work most similar to SR-RNNs are self-clustering RNNs (Zeng et al., 1993). These RNNs learn discretized states, that is, binary valued hidden state vectors, and show that these networks generalize better to longer input sequences. Contrary to self-clustering RNNs, we propose an end-to-end differentiable probabilistic state transition mechanism between cell applications.

Stochastic RNNs are a class of generative recurrent models for sequence data (Bayer & Osendorfer, 2014; Fraccaro et al., 2016; Goyal et al., 2017). They model uncertainty in the hidden states of an RNN by introducing latent variables. Contrary to SR-RNNs, stochastic RNNs do not model probabilistic state transition dynamics. Hence, they do not address the problem of overfitting through hidden state memorization and improvements to DFA extraction.

There are proposals for extending RNNs with various types of external memory. Representative examples are the neural Turing machine (Graves et al., 2014), improvements thereof (Graves et al., 2016), and RNNs augmented with neural stacks, queues, and deques (Grefenstette et al., 2015). Contrary to these proposals, we do not augment RNNs with differentiable data structures but regularize RNNs to make better use of existing memory components such as the cell state. We hope, however, that differentiable neural computers could benefit from state-regularization.

## 6 CONCLUSION

State-regularized RNNs with cell states operate more like automata with external memory and less like DFAs. This results in a markedly improved extrapolation behavior on several datasets. We do not claim, however, that SR-RNNs are a panacea for all problems associated with RNNs. For instance, we could not observe an improved convergence behavior of SR-RNNs. Sometimes SR-RNNs converged faster, sometimes vanilla RNNs. While we have mentioned that the computational overhead of SR-RNNs is modest, it still exists, and this might exacerbate the problem that RNNs often take a long to be trained and tuned. We plan to investigate variants of state regularization and the ways in which it could improve differentiable computers with RNN controllers.

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

## APPENDIX A  PROOFS OF THEOREMS 3.1 AND 3.2

### A.1  THEOREM 3.1

**The state transition behavior of a memory-less SR-RNN using equation 4 is identical to that of a probabilistic finite automaton.**

*Proof.* The state transition function $\delta$ of a probabilistic finite state machine is identical to that of a finite deterministic automaton (see section 2) with the exception that it returns a probability distribution over states. For every state $q$ and every input token $a$ the transition mapping $\delta$ returns a probability distribution $\boldsymbol{\alpha} = (\alpha_1, ..., \alpha_k)$ that assigns a fixed probability to each possible state $q \in \mathcal{Q}$ with $|\mathcal{Q}| = k$. The automaton transitions to the next state according to this distribution. Since by assumption the SR-RNN is using equation 4, we only have to show that the probability distribution over states computed by the stochastic component of a memory-less SR-RNN is identical for every state $q$ and every input token $a$ irrespective of the previous input sequence and corresponding state transition history.

More formally, for every pair of input token sequences $\mathbf{a}_1$ and $\mathbf{a}_2$ with corresponding pair of resulting state sequences $\mathbf{q}_1 = (q_{i_1}, ..., q_{i_n}, q)$ and $\mathbf{q}_2 = (q_{j_1}, ..., q_{j_m}, q)$ in the memory-less SR-RNN, we have to prove, for every token $a \in \Sigma$, that $\boldsymbol{\alpha}_1$ and $\boldsymbol{\alpha}_2$, the probability distributions over the states returned by the stochastic component for state $q$ and input token $a$, are identical. Now, since the RNN is, by assumption, memory-less, we have for both $\mathbf{a}_1, \mathbf{q}_1$ and $\mathbf{a}_2, \mathbf{q}_2$ that the only inputs to the RNN cell are exactly the centroid $\mathbf{s}_q$ corresponding to state $q$ and the vector representation of token $a$. Hence, under the assumption that the parameter weights of the RNN are the same for both state sequences $\mathbf{q}_1$ and $\mathbf{q}_2$, we have that the output $\mathbf{u}$ of the recurrent component (the base RNN cell) is identical for $\mathbf{q}_1$ and $\mathbf{q}_2$. Finally, since by assumption the centroids $\mathbf{s}_1, ..., \mathbf{s}_k$ are fixed, we have that the returned probability distributions $\boldsymbol{\alpha}_1$ and $\boldsymbol{\alpha}_2$ are identical. Hence, the memory-less SR-RNN's transition behavior is identical to that of a probabilistic finite automaton. $\square$

### A.2  THEOREM 3.2

**For $\tau \to 0$ the state transition behavior of a memory-less SR-RNN (using equations 4 or 5) is identical to that of a deterministic finite automaton.**

*Proof.* Let us consider the softmax function with temperature parameter $\tau$

$$\alpha_i = \frac{\exp(b_i/\tau)}{\sum_{i=1}^{k} \exp(b_i/\tau)}$$

for $1 \le i \le k$. SR-RNNs use this softmax function to normalize the scores (from a dot product or Euclidean distance) into a probability distribution. First, we show that for $\tau \to 0^+$, that there is exactly one $M \in \{1, ..., k\}$ such that $\alpha_M = 1$ and $\alpha_i = 0$ for all $i \in \{1, ..., k\}$ with $i \ne M$. Without loss of generality, we assume that there is a $M \in \{1, ..., k\}$ such that $b_M > b_i$ for all $i \in \{1, ..., k\}, i \ne M$. Hence, we can write for $\epsilon_1, ..., \epsilon_k > 0$

$$\alpha_i = \frac{\exp(b_i/\tau)}{\exp((b_M - \epsilon_1)/\tau) + ... + \exp(b_M/\tau) + ... + \exp((b_M - \epsilon_k)/\tau)}$$

$$= \frac{\exp(b_i/\tau)}{\exp(b_M/\tau)\exp(\epsilon_1/\tau)^{-1} + ... + \exp(b_M/\tau) + ... + \exp(b_M/\tau)\exp(\epsilon_k/\tau)^{-1}}$$

$$= \frac{\exp(b_i/\tau)}{\exp(b_M/\tau)\left[\exp(\epsilon_1/\tau)^{-1} + ... + 1 + ... + \exp(\epsilon_k/\tau)^{-1}\right]}.$$

Now, for $\tau \to 0$ we have that $\alpha_M \to 1$ and for all other $i \ne M$ we have that $\alpha_i \to 0$. Hence, the probability distribution $\boldsymbol{\alpha}$ of the SR-RNN is always the one-hot encoding of a particular centroid.

By an argument analog to the one we have made for Theorem 3.1, we can prove that for every state $q \in \mathcal{Q}$ and every input token $a \in \Sigma$, the probability distribution $\boldsymbol{\alpha}$ of the SR-RNN is the same irrespective of the previous input sequences and visited states. Finally, by plugging in the one-hot encoding $\boldsymbol{\alpha}$ in both equations 4 and 5, we can conclude that the transition function of a memory-less SR-RNN is identical to that of a DFA, because we always chose exactly one new state. $\square$

## APPENDIX B  IMPLEMENTATION DETAILS

Unless otherwise indicated we always (a) use single-layer RNNs, (b) learn an embedding for input tokens before feeding it to the RNNs, (c) apply RMSPROP with a learning rate of 0.01 and momentum of 0.9; (d) do not use dropout or batch normalization of any kind; and (e) use state-regularized RNNs based on equations 2&5 with a temperature of $\tau = 1$ (standard softmax). We implemented SR-RNNs with Theano(Theano Development Team, 2016) [2]. The hyper-parameter were tuned to make sure the vanilla RNNs achieves the best performance. For SR-RNNs we tuned the weight initialization values for the centroids and found that sampling uniformly from the interval $[-0.5, 0.5]$ works well across different datasets. Table 9 lists some statistics about the datasets and the experimental set-ups. Table 10 shows the length and nesting depth (if applicable) for the sequences in the train, validation, and test datasets.

| Task | Architecture | Units | $k$ | Train | Valid | Test |
|------|-------------|-------|-----|-------|-------|------|
| Tomita 1 | SR-GRU | 100 | 5, 10, 50 | 265 (12) | 182 (4) | – |
| Tomita 2 | SR-GRU | 100 | 10, 50 | 257 (6) | 180 (2) | – |
| Tomita 3 | SR-GRU | 100 | 50 | 2141 (1028) | 1344 (615) | – |
| Tomita 4 | SR-GRU | 100 | 50 | 2571 (1335) | 2182(1087) | – |
| Tomita 5 | SR-GRU | 100 | 50 | 1651 (771) | 1298(608) | – |
| Tomita 6 | SR-GRU | 100 | 50 | 2523 (1221) | 2222(1098) | – |
| Tomita 7 | SR-GRU | 100 | 50 | 1561 (745) | 680(288) | – |
| BP (large) | SR-LSTM (-P) | 100 | 5 | 22286 (13025) | 6704 (3582) | 1k |
| BP (small) | SR-LSTM (-P) | 100 | 2,5,10,50,100 | 1008 (601) | 268 (142) | 1k |
| Palindrome | SR-LSTM (-P) | 100 | 5 | 229984 (115040) | 50k(25k) | 1k |
| IMDB (full) | SR-LSTM (-P) | 256 | 2,5,10 | 25k | – | 25k |
| IMDB (small) | SR-LSTM (-P) | 256 | 2,5,10 | 25k | – | 25k |
| MNIST (normal) | SR-LSTM (-P) | 256 | 10,50,100 | 60k | – | 10k |
| MNIST (perm.) | SR-LSTM (-P) | 256 | 10,50,100 | 60k | – | 10k |
| Fashion-MNIST | SR-LSTM (-P) | 256 | 10 | 55k | 5k | 10k |
| Wikipedia | SR-LSTM (-P) | 300 | 1000 | 22.5M | 1.2M | 1.2M |

Table 9: A summary of dataset and experiment characteristics. The values in parentheses are the number of positive sequences.

| Task | Train $l$ & $d$ | Valid $l$ & $d$ | Test $l$ & $d$ |
|------|-----------------|------------------|-----------------|
| Tomita 1 -7 | $l = 0 \sim 13, 16, 19, 22$ | $l = 1, 4,..., 28$ | – |
| BP (large) | $d \in [1, 5]$ | $d \in [6, 10]$ | $d \in [1, 20]$ |
| BP (small) | $d \in [1, 5]$ | $d \in [6, 10]$ | $d \in [1, 20]$ |
| Palindrome | $l \in [1, 25]$ | $l \in [26, 50]$ | $l \in [50, 500]$ |
| IMDB (full) | $l \in [11, 2820], l_{aver} = 285$ | – | $l \in [8, 2956], l_{aver} = 278$ |
| IMDB (small) | $l = 10$ | – | $l \in [100, 200]$ |
| MNIST(normal) | $l = 784$ | $l = 784$ | $l = 784$ |
| MNIST(perm.) | $l = 784$ | $l = 784$ | $l = 784$ |
| Fashion-MNIST | $l = 784$ | $l = 784$ | $l = 784$ |

Table 10: The lengths ($l$) and depths ($d$) of the sequences in the training, validation, and test sets of the various tasks.

## APPENDIX C  ADDITIONAL EXPERIMENTS AFTER INITIAL REVIEWS

We conducted several additional experiments based on the feedback from the ICLR reviewers.

### C.1  LSTM-P BASELINE ON PALINDROME, BP AND IMDB

The results of LSTM-P on IMDB (full) and sequential MNIST are listed in Table 5 and Table 7. Here we report its performance on the Palindrome dataset, the BP dataset (large) and IMDB (small).

---

[2]http://www.deeplearning.net/software/theano/

The results for LSTM-P on the Palindrome dataset, BP (large) and IMDB (small) are reported in Table 11, Table 12 and Table 13 respectively. We observe that the LSTM-P achieves results similar to that of the LSTM.

| max length | 100 | 200 | 500 |
|---|---|---|---|
| LSTM | 31.2 | 42.0 | 47.7 |
| LSTM-P | 28.4 | 36.2 | 41.5 |
| SR-LSTM | 28.0 | 36.0 | 44.6 |
| SR-LSTM-P | **10.5** | **16.7** | **29.8** |

Table 11: Error rates in % on sequences of varying lengths from the Palindrome test set.

| Dataset Models | Large Dataset | | | |
|---|---|---|---|---|
| | LSTM | SR-LSTM | LSTM-P | SR-LSTM-P |
| $d \in [1, 10], l \leq 100$ | 0.005 | 0.038 | 0.022 | **0.000** |
| $d \in [10, 20], l \leq 100$ | 0.334 | 0.255 | 0.308 | **0.001** |
| $d \in [10, 20], l \leq 200$ | 0.341 | 0.313 | 0.351 | **0.003** |
| $d = 5, l \leq 200$ | 0.002 | 0.044 | 0.013 | **0.000** |
| $d = 10, l \leq 200$ | 0.207 | 0.227 | 0.254 | **0.004** |
| $d = 20, l \leq 1000$ | 0.543 | 0.540 | 0.496 | **0.020** |

Table 12: Error rates for the balanced parentheses (BP) test sets ($d$=depth, $l$=length, $k = 5$).

| model length | LSTM | | SR-LSTM | | LSTM-P | | SR-LSTM-P | |
|---|---|---|---|---|---|---|---|---|
| | 100 | 200 | 100 | 200 | 100 | 200 | 100 | 200 |
| train error | 29.23 | 32.77 | 27.17 | 31.72 | 25.50 | 29.05 | **23.30** | **24.67** |
| test error | 29.96 | 33.19 | 28.82 | 32.88 | 28.02 | 30.12 | **26.04** | **26.21** |
| time | 400.29s/epoch | | 429.15s/epoch | | 410.29s/epoch | | 466.10s/epoch | |

Table 13: Error rates (on training and test splits of the IMDB data) in % and averaged training time in seconds when training only on truncated sequences of length 10.

## C.2 LANGUAGE MODELING

We evaluated the SR-LSTMs on language modeling task with the Wikipedia dataset(Daniluk et al., 2017b)[3]. It consists of 7500 English Wikipedia articles. We used the same experimental setup as in previous work (Daniluk et al., 2017b). We used the provided training, validation, and test dataset: 22.5M words in the training set, 1.2M in the validation, and 1.2M words in the test set. We used the 77k most frequent words from the training set as vocabulary. We report the results in Table 14. We use the model with the best perplexity on the on the validation set. Note that we only tuned the number of centroids for SR-LSTM and SR-LSTM-P and used the same hyperparameters that were used for the vanilla LSTMs.

The results show that the perplexity results for the SR-LSTM and SR-LSTM-P outperform those of the vanilla LSTM and LSTM with peephole connection. The difference, however, is modest and we conjecture that the ability to model long-range dependencies is not that important for this type of language modeling tasks. This is an observation that has also been made by previous work Daniluk et al. (2017b). The perplexity of the SR-LSTMs cannot reach that of state of the art methods. The methods, however, all utilize a mechanism (such as attention) that allows the next-word decision to be based on a number of past hidden states. The SR-LSTMs, in contrast, makes the next-word decision only based on the current hidden state.

---

[3]The wikipedia corpus is available at https://goo.gl/s8cyYa

| Model | $a$ | $\theta_{W+M}$ | $\theta_M$ | Dev | Test |
|---|---|---|---|---|---|
| RNN | - | 47.0M | 23.9M | 121.7 | 125.7 |
| LSTM | - | 47.0M | 23.9M | 83.2 | 85.2 |
| FOFE HORNN(3-rd order)(Soltani & Jiang, 2016) | - | 47.0M | 23.9M | 116.7 | 120.5 |
| Gated HORNN(3-rd order)(Soltani & Jiang, 2016) | - | 47.0M | 23.9M | 93.9 | 97.1 |
| RM(+tM-g) (Tran et al., 2016) | 15 | 93.7M | 70.6M | 78.2 | 80.1 |
| Attention (Daniluk et al., 2017b) | 10 | 47.0M | 23.9M | 80.6 | 82.0 |
| Key-Value (Daniluk et al., 2017b) | 10 | 47.0M | 23.9M | 77.1 | 78.2 |
| Key-Value Predict (Daniluk et al., 2017b) | 5 | 47.0M | 23.9M | 74.2 | 75.8 |
| 4-gram RNN (Daniluk et al., 2017b) | - | 47.0M | 23.9M | 74.8 | 75.9 |
| LSTM-P | - | 47.0M | 23.9M | 85.8 | 86.9 |
| SR-LSTM ($k = 1000$) | - | 47.3M | 24.2M | 81.8 | 83.0 |
| SR-LSTM-P ($k = 1000$) | - | 47.3M | 24.2M | 82.3 | 84.5 |

Table 14: The perplexity results for the SR-LSTMs and state of the art methods. Here, $\theta_{W+M}$ are the number of model parameters and $\theta_M$ the number of model parameters without word representations. $a$ is the attention window size.

## APPENDIX D   TOMITA GRAMMARS AND DFA EXTRACTION

The Tomita grammars are a collection of 7 regular languages over the alphabet {0,1} (Tomita, 1982). Table 15 lists the regular grammars defining the Tomita grammars.

| Grammars | Descriptions |
|---|---|
| 1 | 1* |
| 2 | (10)* |
| 3 | An odd number of consecutive 1s is followed by an even number of consecutive 0s |
| 4 | Strings not contain a substring "000" |
| 5 | The numbers of 1s and 0s are even |
| 6 | The difference of the numbers of 1s and 0s is a multiple of 3 |
| 7 | 0*1*0*1* |

Table 15: The seven Tomita grammars (Tomita, 1982).

We follow previous work (Wang et al., 2018b; Schellhammer et al., 1998) to construct the transition function of the DFA (deterministic finite automata).

---

**Algorithm 1:** Computes DFA transition function

---

**Input:** pre-trained SR-RNN, dataset $\mathbf{D}$, alphabet $\Sigma$, start token $\mathtt{s}$
**Output:** transition function $\delta$ of the DFA
**for** $i, j \in \{1, ..., k\}$ *and all* $x \in \Sigma$ **do**
  $\quad \mathcal{T}[(c_i, x_t, c_j)] = 0$             # initialize transition counts to zero
$\{p_i\}_{i=1}^k \leftarrow$ SR-RNN($\mathtt{s}$)        # compute the transition probabilities for the start token
$j = \arg\max_{i \in \{1,...,k\}}(p_i)$      # determine $j$ the centroid with max transition probability
$c_0 = j$                 # set the start centroid $c_0$ to $j$
**for** $\mathbf{x} = (x_1, x_2, ..., x_T) \in \mathbf{D}$ **do**
  $\quad$ **for** $t \in [1, ..., T]$ **do**
    $\quad\quad \{p_j\}_{j=1}^k \leftarrow$ SR-RNN($x_t$)      # compute the transition probabilities for the $t$-th token
    $\quad\quad j = \arg\max_{i \in \{1,...,k\}}(p_i)$    # determine $j$ the centroid with max transition probability
    $\quad\quad c_t = j$             # set $c_t$, the centroid in time step $t$, to $j$
    $\quad\quad \mathcal{T}[(c_{t-1}, x_t, c_t)] \leftarrow \mathcal{T}[(c_{t-1}, x_t, c_t)] + 1$      # increment transition count
**for** $i \in \{1, ..., k\}$ *and* $x \in \Sigma$ **do**
  $\quad \delta(i, x) = \arg\max_{j \in \{1,...,k\}} \mathcal{T}[(i, x, j)]$      # compute the transition function of the DFA
**return** $\delta$

---

We follow earlier work (Weiss et al., 2018b) and attempt to train a GRU to reach 100% accuracy for both training and validation data. We first trained a single-layer GRU with 100 units on the

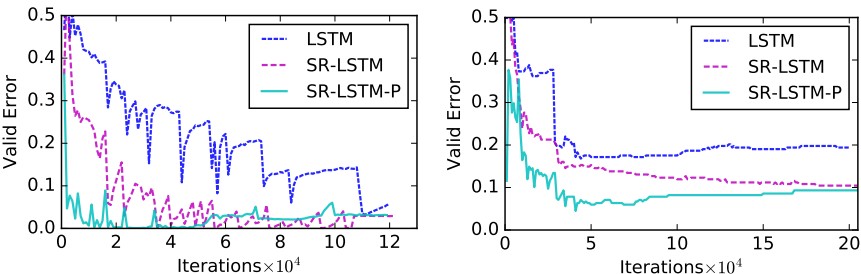

Figure 8: The DFA extracted from the SR-GRU for Tomita grammar 7. The numbers inside the circles correspond to the centroid indices of the SR-GRU. Double circles indicate accept states.

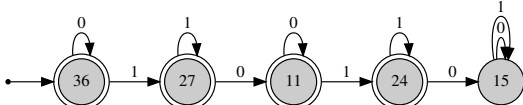

Figure 9: The error curves on the validation data for the LSTM, SR-LSTM, and SR-LSTM-P ($k = 5$)on the large BP dataset (left) and the small BP dataset (right).

data. We use GRUs since they are memory-less. Whenever the GRU converged within 1 hour to a training accuracy of $100\%$, we also trained a SR-GRU based on equations 2&5 with $k = 50$ and $\tau = 1$. This was the case for the grammars 1-4 and 7. For grammar 5 and 6, our experiments show that both vanilla GRU and SR-GRU were not able to achieve $100\%$ accuracy. In this case, SR-GRU ($97.2\%$ train and $96.8\%$ valid accuracy) could not extract the correct DFA for grammar 5 and 6. The exploration of deeper GRUs and their corresponding SR-GRUs (2 layers as in (Weiss et al., 2018b) for DFA extraction could be interesting future work.

Algorithm 1 lists the pseudo-code of the algorithm that constructs the transition function of the DFA. Figure 8 shows the extracted DFA for grammar 7. All DFA visualization in the paper are created with GraphViz [4].

## APPENDIX E   TRAINING CURVES

Figure 9 plots the validation error during training of the LSTM, SR-LSTM, and SR-LSTM-P on the BP (balanced parentheses) datasets. Here, the SR-LSTM and SR-LSTM both have $k = 5$ centroids. The state-regularized LSTMs tend to reach better error rates in a shorter amount of iteration.

Figure 10 (left) plots the test error of the LSTM, SR-LSTM, and SR-LSTM-P on the IMDB dataset for sentiment analysis. Here, the SR-LSTM and SR-LSTM both have $k = 10$ centroids. In contrast to the LSTM, both the SR-LSTM and the SR-LSTM do not overfit.

Figure 10 (right) plots the test error of the SR-LSTM-P when using either (a) the last hidden state and (b) the cell state as input to the classification function. As expected, the cell state contains also valuable information for the classification decision. In fact, for the SR-LSTM-P it contains more information about whether an input sequence should be classified as positive or negative.

## APPENDIX F   VISUALIZATION, INTERPRETATION, AND EXPLANATION

The stochastic component and its modeling of transition probabilities and the availability of the centroids facilitates novel ways of visualizing and understanding the working of SR-RNNs.

---

[4]https://www.graphviz.org/

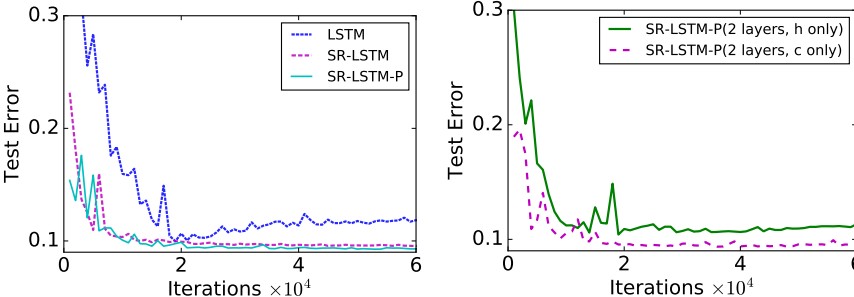

Figure 10: The error curves on the test data for the LSTM, SR-LSTM, SR-LSTM-P ($k = 10$) on the IMDB sentiment analysis dataset. (Left) It shows state-regularized RNNs show better generalization ability. (Right) A 2-layer SR-LSTM-P achieves better error rates when the classification function only looks at the last cell state compared to it only looking at the last hidden state.

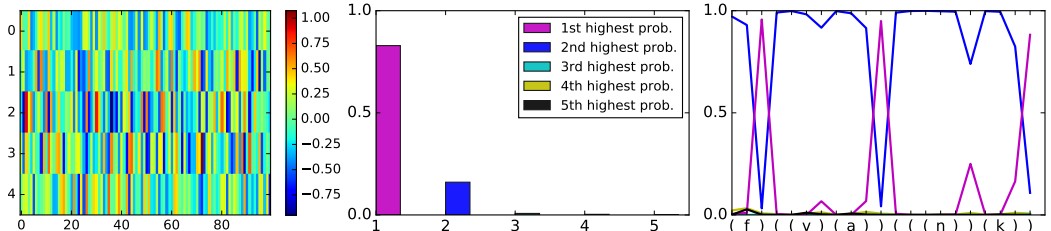

Figure 11: Visualization of a SR-LSTM-P with $k = 5$ centroids, a hidden state dimension of 100, trained on the large BP data. (Left) visualization of the learned centroids. (Center) mean transition probabilities when ranked highest to lowest. This shows that the transition probabilities are quite spiky. (Right) transition probabilities for a specific input sequence.

## F.1 BALANCED PARENTHESES

Figure 11 (left) shows the $k = 5$ learned centroids of a SR-LSTM-P with hidden state dimension 100.

Figure 11 (center) depicts the average of the ranked transition probabilities for a large number of input sequences. This shows that, on average, the transition probabilities are spiky, with the highest transition probability being on average $0.83$, the second highest $0.16$ and so on.

Figure 11 (right) plots the transition probabilities for a SR-LSTM-P with $k = 5$ states and hidden state dimension 100 for a specific input sequence of BP.

Figure 12 visualizes the hidden states $\mathbf{h}$ of a LSTM, SR-LSTM, and SR-LSTM-P trained on the large BP dataset. The SR-LSTM and SR-LSTM-P have $k = 5$ centroids and a hidden state dimension of 100. One can see that the LSTM memorizes with its hidden states. The evolution of its hidden states is highly irregular. The SR-LSTM and SR-LSTM-P, on the other hand, have a much more regular behavior. The SR-LSTM-P utilizes mainly two states to accept and reject an input sequence.

## F.2 SENTIMENT ANALYSIS

Since we can compute transition probabilities in each time step of an input sequence, we can use these probabilities to generate and visualize prototypical input tokens and the way that they are associated with certain centroids. We show in the following that it is possible to associate input tokens (here: words of reviews) to centroids by using their transition probabilities.

For the sentiment analysis data (IMDB) we can associate words to centroids for which they have high transition probabilities. To test this, we fed all test samples to a trained SR-LSTM-P. We

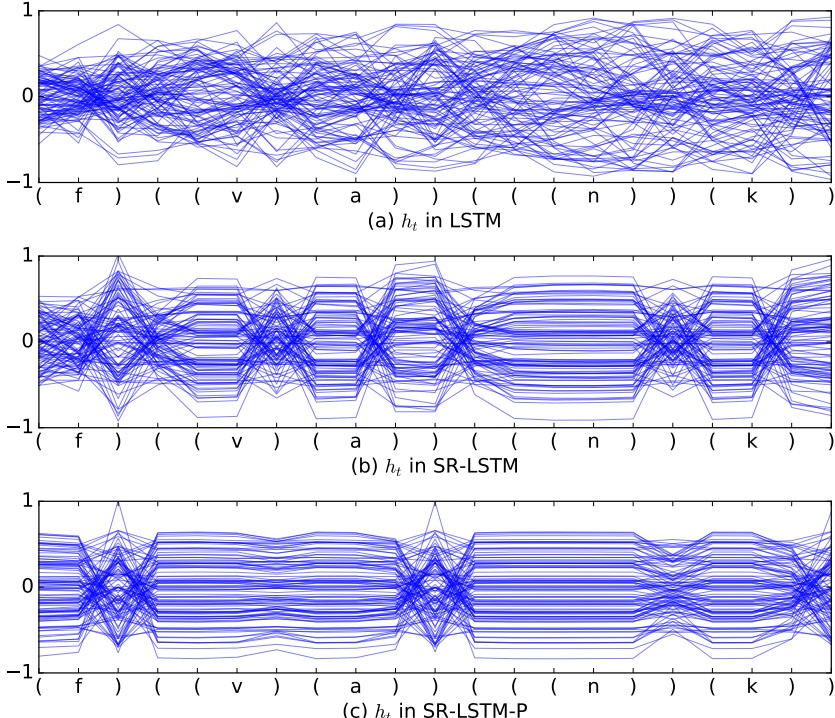

Figure 12: Visualizations of the hidden states **h** of a vanilla LSTM, an SR-LSTM, and a SR-LSTM-P ($k = 10$) for a specific input sequence. All models were trained on BP and have hidden state dimension of $100$. The LSTM memorizes with its hidden state. The SR-LSTM and SR-LSTM-P utilize few states and have a more stable behavior over time.

| Centroids | Top-5 words with probabilities |
|---|---|
| centroid 0 | piece (0.465) instead (0.453) slow (0.453) surface (0.443) artificial (0.37) |
| centroid 1 | told (0.752) mr. (0.647) their (0.616) she (0.584) though (0.561) |
| centroid 2 | just (0.943) absolutely (0.781) extremely (0.708) general (0.663) sitting (0.587) |
| centroid 3 | worst (1.0) bad (1.0) pointless (1.0) boring (1.0) poorly (1.0) |
| centroid 4 | jean (0.449) bug (0.406) mind (0.399) start (0.398) league (0.386) |
| centroid 5 | not (0.997) never (0.995) might (0.982) at (0.965) had (0.962) |
| centroid 6 | against (0.402) david (0.376) to (0.376) saying (0.357) wave (0.349) |
| centroid 7 | simply (0.961) totally (0.805) c (0.703) once (0.656) simon (0.634) |
| centroid 8 | 10 (0.994) best (0.992) loved (0.99) 8 (0.987) highly (0.987) |
| centroid 9 | you (0.799) strong (0.735) magnificent (0.726) 30 (0.714) honest (0.69) |

Table 16: List of prototypical words for the $k = 10$ centroids of an SR-LSTM-P trained on the IMDB dataset. The top-5 highest transition probability words are listed for each centroid. We colored the positive centroid words in green and the negative centroid words in red.

determine the average transition probabilities for each word and centroid and select those words with the highest average transition probability to a centroid as the prototypical words of said centroid. Table 16 lists the top 5 words according to the transition probabilities to each of the 5 centroids for the SR-LSTM-P with $k = 10$. It is now possible to inspect the words of each of the centroids to understand more about the working of the SR-RNN.

Figure 14 and 15 demonstrate that it is possible to visualize the transition probabilities for each input sequence. Here, we can see the transition probabilities for one positive and one negative sentence for a SR-LSTM-P with $k = 5$ centroids.

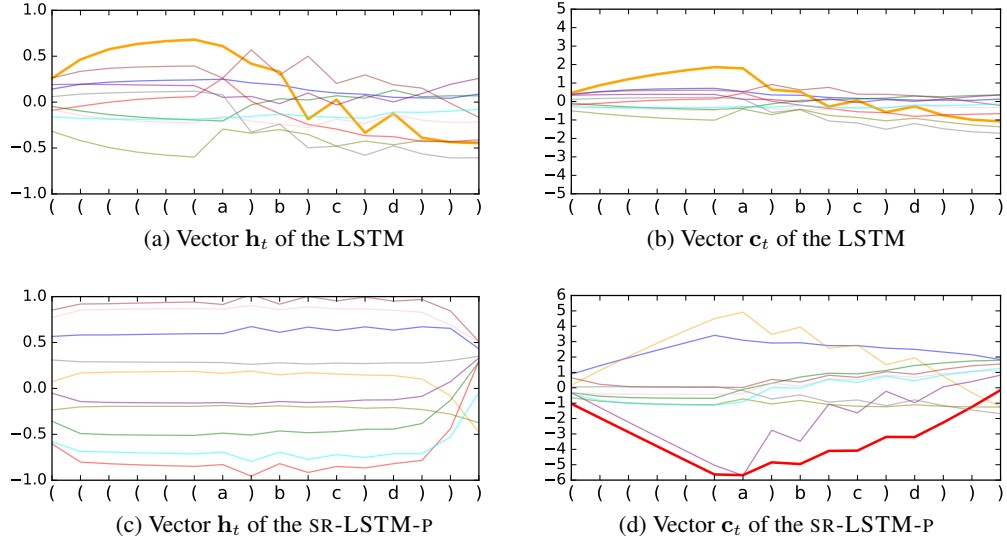

Figure 13: Visualization of hidden state $\mathbf{h}_t$ and cell state $\mathbf{c}_t$ of the LSTM and the SR-LSTM-P for a specific input sequence from BP. The LSTM memorizes the number of open parentheses both in the hidden and to a lesser extent in the cell state (bold yellow lines). The memorization is not accomplished with saturated gate outputs and a drift is observable for both vectors. The SR-LSTM-P maintains two distinct hidden states (accept and reject) and does not visibly memorize counts through its hidden states. The cell state is used to cleanly memorize the number of open parentheses (bold red line) with saturated gate outputs ($\pm 1$). A state vector drift is not observable.

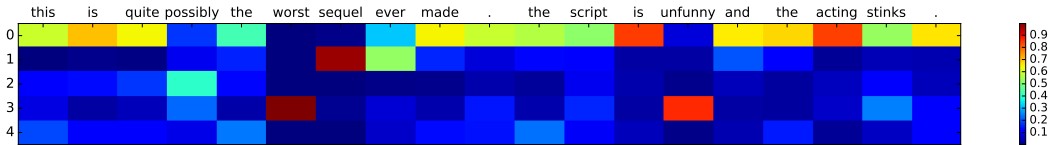

Figure 14: Visualization of the transition probabilities for an SR-LSTM-P with $k = 5$ centroids trained on the IMDB dataset for a negative input sentence.

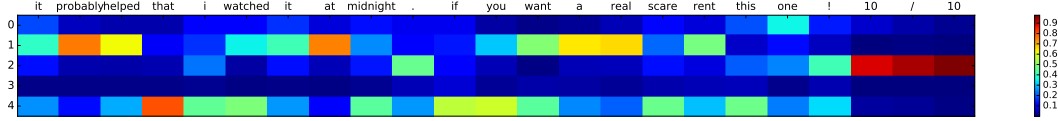

Figure 15: Visualization of the transition probabilities for an SR-LSTM-P with $k = 5$ centroids trained on the IMDB dataset for a positive input sentence.

### F.3   MNIST AND FASHION-MNIST

For pixel-by-pixel sequences, we can use the SR-RNNs to directly generate prototypes that might assist in understanding the way SR-RNNs work. We can compute and visualize the average transition probabilities for all examples of a given class. Note that this is different to previous post-hoc methods (e.g., activation maximization (Berkes & Wiskott, 2006; Nguyen et al., 2016)), in which a network is trained first and in a second step a second neural network is trained to generate the prototypes. Figure 16 visualizes the prototypes (average transition probabilities for all examples from a digit class) of SR-LSTM-P for $k = 10$ centroids. One can see that each centroid is paying attention to a different part of the image.

In addition, SR-RNN can be used to visualize the transition probabilities for specific inputs. To explore this, we trained an SR-LSTM-P ($k = 10$) on the MNIST (accuracy 98%) and Fashion

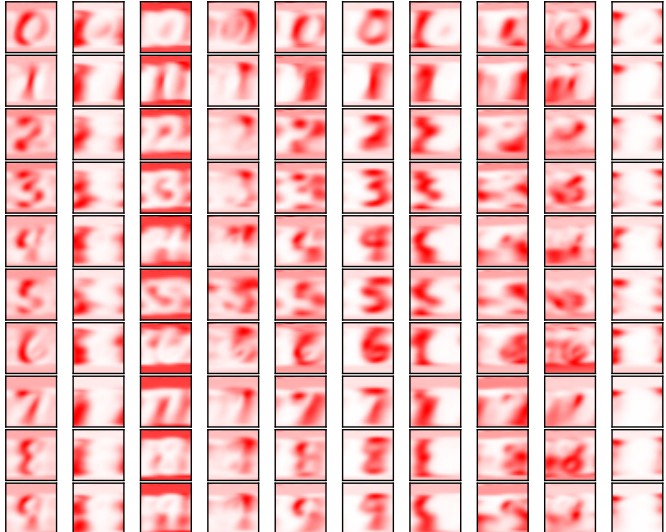

Figure 16: Visualization of average transition probabilities of the SR-LSTM-P with $k = 10$ centroids, over all test images. Each row represents a digit class (a concept) and each column depicts the prototype (average transition probability) for each of the centroids.

MNIST (86%) data (Xiao et al., 2017), having the models process the images pixel-by-pixel as a large sequence. Figure 17 visualizes the transition probabilities with a heatmap for specific input images.

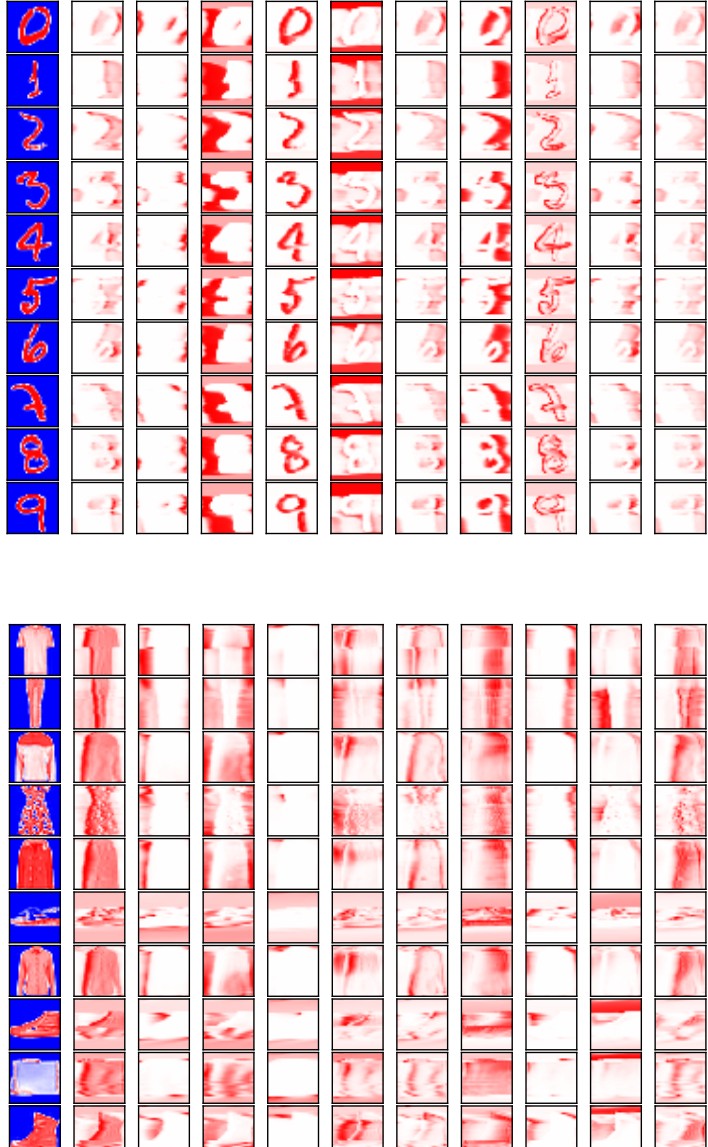

Figure 17: We can visualize the working of an SR-LSTM-P on a specific input image by visualizing the transition probabilities of each of the centroids (here: $k = 10$). (Top) The visualizetion for some MNIST images. (Bottom) The visualization for some Fashion-MNIST images. The first column depicts the input image and the second to to $11^{th}$ the state transition probability heatmaps corresponding to the 10 centroids.

