# OpenReview forum: "State-Regularized Recurrent Networks"
_ICLR.cc/2019/Conference_

### Official Review · AnonReviewer2 · 2018-11-02
**Interesting idea, but shines only on specifically designed benchmarks, needs more experiments on well established datasets**

**Rating:** 5
**Confidence:** 5

**Review:**

The paper proposes an RNN architecture inspired from deterministic pushdown automata. An RNN is extended to use soft attention at every time step to choose from several learnable centroids.

In general, the paper is well written and the proposed model is theoretically grounded. Unfortunately, the proposed approach shines only on specifically designed benchmarks. It is not a surprise that a CF can be learned by an architecture very similar to DPDA (with addition of learnable parameters). There is a number of specifically designed tasks to test long-term memorization, such as copy/addition, etc. Furthermore, RNNs are mostly used for natural language processing tasks. This paper only conducts experiments on IMDB sentiment analysis ignoring better benchmarked tasks, such as language modelling.

It is not absolutely clear why authors claim that cell is playing the role of memory. It is always possible to rewrite LSTM formulas with h' which is concatenation of hidden state h and cell c. Results on "peephole connection"-inspired SR-LSTM-p should be benchmarked against an LSTM with peephole connections.

The claim repeated several times that RNNs operate like DFAs, not DPDAs. This is an important point in the paper and should be verbalized more. Does it mean that it is easier to learn regular languages with RNNs?

While intuitive, theorems 3.1-3.2 are very vague to be theorems. Otherwise, they should be proven or provided a sketch of proof. For example, how do you formalize "state dynamics"?

The quality of writing of the related work section is worse that the rest of the paper. Authors should explore more other hidden state regularization methods. And, perhaps, give less attention to stochastic RNNs since the final version of the proposed model is not stochastic.

To summarize, this paper provides an interesting direction but lacks in terms of experimentation and global coherence of what is claimed and what is shown.

Minor points:
- Citation of Theano is missing
- Give a sentence explaining what is hidden state "drifting"
- a-priori -> a priori

---

> ### Author Response · Authors · 2018-11-10
> **Response to AnonReviewer2**
>
> Thank you very much for your helpful review.
>
> Due to your remarks, we have conducted additional experiments on a commonly used language modeling dataset. These results can be found in Appendix C. We want to emphasize that, while we also outperform the vanilla LSTMs on said dataset, a number of previous papers (e.g., [1]) have shown that the ability to model long-range dependencies is not so important in language modeling (i.e., next word prediction). A better way to handle long-range dependencies can improve the results only modestly since the majority of predictions can be accurately made based on a few previous words. It is for this reason that prior work has often used IMDB and sequential MNIST because the ability to model long-range dependencies is more crucial for these datasets. Please also note that we see our paper in line with previous work on DFA extraction [2, 3] which is why we chose the BP and Tomita datasets. Moreover, the Palindrome language we use in the paper is similar to the Copy problem as it is also required to memorize all the input tokens. It is therefore very similar to the copy task that you mention in your review. The results in Table 3 show that the sr-lstm-p outperforms the vanilla LSTM on the Palindrome data.
>
> Previous work has shown that LSTMs are better than GRUs at counting due to their ability to memorize counts in the cell state. For instance, in [2] it was shown that LSTMs perform much better than GRUs on simpler counting tasks (simpler than e.g. the BP language we use).  In Figure 6, 12, and 13 we also show that the cell state is indeed used as memory in the sr-lstm-p. Hence, there is evidence that indeed, the cell state is used as memory in LSTMs. With our work, we want to encourage RNNs to mainly use the cell state for memorization.
>
> We have added additional results (Appendix C) for the LSTM with peephole connections and without state-regularization. We agree that this is important. We show that the sr-lstm-p outperforms the lstm-p.
>
> Based on your helpful input, we have also included a more thorough description of what we mean when we say that vanilla RNNs often operate more like DFAs. In short, DFAs memorize everything about the current input sequence in the current state and do not have access to additional memory components. RNNs (without state regularization) tend to do the same based on empirical results. Indeed, in Figure 6 we show that most of the memorization is accomplished with the hidden state and to a lesser extent by the cell state. There is also evidence from previous work. For instance, in [3] was shown that LSTMs trained on BP behaved more like DFAs.
>
> We fully agree with your comment that Theorems 3.1 and 3.2 were vaguely formulated. We have improved the formulation of the theorems and added proofs of both theorems (new Appendix A).
>
> We have also included a few more citations in the related work section. We would highly appreciate pointers to additional LSTM regularization methods that we could add to the discussion.
>
> We have fixed all the issues you have listed under “minor points.” We have added a citation to Theano, we have fixed the typo, and added a sentence describing the state drift behavior first observed in [4].
>
> Finally, we want to emphasize that the main contribution we see in state-regularized LSTMs is to gain a better understanding of the workings of RNNs and the ways one could encourage them to behave more like automata with external memory. It is not our intention to show that these models outperform all state of the art methods. We have put more emphasis on understanding what and how exactly RNNs learn (by e.g. creating visualizations such as Figure 6) and less on tuning them to outperform the state of the art. Please note that this is an up and coming research theme with the recent BlackboxNLP (https://blackboxnlp.github.io/) workshop having more than 600 attendees.
>
> Thank you again for your helpful review. We hope you take our response and revised paper into account. Please let us know if there is anything else we can do to improve the paper.
>
>
> [1] Michał Daniluk, Tim Rocktäschel, Johannes Welbl, and Sebastian Riedel. Frustratingly short attention spans in neural language modeling. 2017b.
> [2] Gail Weiss, Yoav Goldberg, and Eran Yahav. On the practical computational power of finite precision rnns for language recognition. 2018.
> [3] Gail Weiss, Yoav Goldberg, and Eran Yahav. Extracting automata from recurrent neural networks using queries and counterexamples. 2018.
> [4] Zheng Zeng, Rodney M Goodman, and Padhraic Smyth. Learning finite state machines with self-clustering recurrent networks. 1993.

---

### Official Review · AnonReviewer3 · 2018-11-08
**Good paper on regularizing the hidden state of LSTM to make sure it uses the cell state properly.**

**Rating:** 6
**Confidence:** 5

**Review:**


Summary:

This paper is based on the observation that LSTMs use the hidden state to memorize information and the cell state (memory) is not fully utilized. To encourage the LSTM to utilize the cell state, authors constraint the hidden state to a set of centroid states and learn to transition between these centroids in a soft way. Authors demonstrate their model in learning simple regular and context-free languages and also in a couple of non-synthetic tasks. The proposed model also has some interpretability of internal state transitions.

Major comments:

1.	The main claim of the paper is that SR-LSTM can extrapolate to longer sequences, unlike LSTM. However, the sequence lengths considered are too small. It would be interesting to train both models with specific sequence length and then keep testing them with longer sequence length and compare the performance. If SR-LSTM behaves like a DPDA, then with larger cell state, the performance should not drop as you increase the sequence length till the capacity of the cell state.

2.	Theorem 3.1 and 3.2 have no proofs. Please make them as notes rather than theorems.

3.	What do different colors in Figure 6 stands for?

4.	In the MNIST task authors claim that they have significant improvement when compared to LSTM. I am not sure if that is accurate. Also, why do you compare SR-LSTM-p only with LSTM? What is the performance of LSTM-p? Please report that as well.

5.	Even in table 3, can you please report the performance of LSTM-p?

Even though the paper does not show strong empirical performance in real-world tasks, I would still recommend for accepting this paper for its contributions in understanding RNNs better, provided authors answer to question 1, 4, and 5.


Minor comments:

1.	Fig 6 is not referred anywhere.

---

> ### Author Response · Authors · 2018-11-14
> **Response to AnonReviewer3**
>
> Thank you very much for the encouraging and helpful review.
>
> First of all, all points you are raising are good observations and have helped us to improve the paper. We have uploaded an additional revised version.
>
> Let us address your concerns one by one.
>
>
> Comment 1: "It would be interesting to train both models with specific sequence length and then keep testing them with longer sequence length and compare the performance.”
>
> If we don’t misunderstand your suggestion, this is what we did with the experiments using the BP (Table 1), Palindrome (Table 3), and IMDB data (Table 4). In all of these cases, we have trained the model on shorter sequences (or sequences with smaller nesting depth as in BP), and then evaluated the trained models on longer sequences. With these experiments, we want to show that state-regularized LSTMs with peephole connections tend to extrapolate better to more complex/longer sequences even when trained on shorter sequences only. Please let us know if we misunderstood your comment.
>
>
> Comment 2: “Theorem 3.1 and 3.2 have no proofs. Please make them as notes rather than theorems.”
>
> We completely agree and have updated both the statements of the theorems and added detailed proofs to the appendix (new Appendix A).
>
>
> Comment 3: “What do different colors in Figure 6 stands for?”
>
> Each color corresponds to one dimension in the respective state vectors (hidden or cell state). In Figure 6, we visualize the hidden and cell states each having 10 dimensions. For instance, the red line in Figure 6(d) was highlighted by us to show that this particular value of the cell state vector of the SR-LSTM-p memorizes the number of encountered open parentheses.
>
>
> Comment 4: “In the MNIST task authors claim that they have significant improvement when compared to LSTM. I am not sure if that is accurate. Also, why do you compare SR-LSTM-p only with LSTM? What is the performance of LSTM-p? Please report that as well.”
>
> The SR-LSTM-p leads to an accuracy improvement of 1.5 percentage points on sequential MNIST and 0.9 percentage points on permuted MNIST. These differences have been considered significant in previous work. However, we are not insisting on the word “significant” and are happy to use simply “improvement.” We have changed this in the revised version.
>
> Comment 5: “Even in table 3, can you please report the performance of LSTM-p?”
>
> Based on your comments (4) and (5), we have included additional experimental results for the LSTM-p. We report results of the LSTM-p on all datasets including IMDB and MNIST. Moreover, we have conducted an additional set of experiments on a language modeling dataset and have included this in Appendix C. All LSTM-p results have either been added to the tables in the main part of the paper or have been added to the said appendix.
>
>
> Minor comment: “Fig 6 is not referred anywhere.”
>
> Thank you. We have fixed the reference to the Figure in the paper.
>
>
> We want to thank you for acknowledging the merit of our contribution as being in line with work that attempts to better understand what and how RNNs learn. The main aim of our work is not to tune state-regularized LSTMs to outperform state of the art methods on benchmark datasets. Rather, we want to understand and develop a mechanism that allows one to encourage RNNs to operate more like automata with external memory. That’s why we have focused on visualizations such as Figure 6-8. Moreover, we believe that the proposed method allows one to extract DFAs representing the state transition behavior and supports novel ways to interpret the working of RNNs on text and visual data.
>
> Again, thank you. Please let us know if there is anything else we can do to improve the paper.

---

### Official Review · AnonReviewer4 · 2018-11-08
**Interesting approach on embedding finite state space transitions into RNN dynamics. General empirical usefulness remains to be seen.**

**Rating:** 6
**Confidence:** 4

**Review:**

This paper proposes a novel architecture and regularization technique for RNN, where the hidden state of an RNN is one of (or a soft weighted average of) a finite number of learnable clusters. This has two claimed benefits: (1) extracting finite state automata from an RNN is much simpler, and (2) forces RNN to operate like an automata and less like finite state machines. The authors make (1) immediately clear, and show (2) with empirical results.

Major comments:

(1) No experiments on widely used benchmarks for RNNs (e.g. language modeling, arithmetic tasks (for instance see Zaremba and Sutskever, 2015) ). Have you tried this by any chance?

(2) Theorems 3.1 and 3.2 are presented without proof. Will be good to at least include it in the appendix.

(3) IMDB experiments: you claim that SR-LSTM and SR-LSTM-p have "superior" extrapolation capabilities than vanilla LSTMs. However, as SR-LSTM and SR-LSTM-p give far lower train error rate, it's not strictly fair to claim that they extrapolate better to longer sequences than encountered during training time.

Is the number of parameters held constant across 3 models? I'm struggling to understand why the training performance of the proposed models is significantly better than pure LSTM. For SR-LSTM-P I can see this being the case (the peephole connections effectively increase the hidden state size), but why does SR-LSTM (whose hidden states should be more constrained than pure LSTMS) perform better than LSTM during training? This makes me wonder whether SR-LSTM and SR-LSTM-P have higher capacity than LSTM somehow.

(4) MNIST experiments : please include results for SR-LSTM

Minor comments:

(1) page 8 : MNIST imagse -> images

---

> ### Author Response · Authors · 2018-11-14
> **Response to AnonReviewer4**
>
> Thank you very much for your insightful review.
>
> We have uploaded a revision of the paper which we hope addresses all of your concerns.
>
> Let us address your comments and questions one by one.
>
>
> Comment (1): “No experiments on widely used benchmarks for RNNs (e.g. language modeling, arithmetic tasks (for instance see Zaremba and Sutskever, 2015) ). Have you tried this by any chance?”
>
> Sequential MNIST and IMDB are benchmarks commonly used for RNNs, specifically when assessing their memorization behavior. However, based on your comment and the comments of your fellow reviewers, we have added experiments for a commonly used language modeling dataset. We have added the results in Appendix C. In short, we outperform the vanilla LSTM and the LSTM with peephole connections. However, we cannot outperform the state of the art on this dataset which typically uses some form of attention mechanism. Please note that our objective is not primarily to outperform the state of the art on all benchmark datasets. Rather, we want to better understand the ways in which we can interpret and improve the learning behavior of RNNs. Previous work with this motivation has often not conducted any experiments on non-synthetic datasets (e.g., [2]).
>
> In [1] the task used to evaluate memorization capabilities of LSTMs is the “Copy task” which was also mentioned by reviewer 2. Please note that the Palindrome task (recognizing ww^{-1}) also requires memorization of the entire sequence w. Based on your comment, we have included a reference to [1] in the paper. Also note that, similar to [1,2], we have used curriculum learning in the experiments for BP and Palindrome for all methods to have a fair comparison.
>
>
> Comment (2): “Theorems 3.1 and 3.2 are presented without proof. Will be good to at least include it in the appendix.”
>
> We have included full proofs in a new appendix (Appendix A).
>
>
> Comment (3): “IMDB experiments: you claim that SR-LSTM and SR-LSTM-p have "superior" extrapolation capabilities than vanilla LSTMs. However, as SR-LSTM and SR-LSTM-p give far lower train error rate, it's not strictly fair to claim that they extrapolate better to longer sequences than encountered during training time.”
>
> Thanks for pointing out that we left enough room for misunderstanding the results. The error rates listed in Table 4 are for sequences of length 100 and 200 when training on truncated sequences of length 10. Hence, we trained the models on truncated sequences of length 10 and tested the models on the training data (with length 100 and 200) and the test data (with length 100 and 200). Hence, the training error is not the error on the truncated sequences of length 10. We hope that this clarifies the experimental results listed in Table 4.
>
> While it is true that the SR-LSTM and SR-LSTM-p have more parameters than the LSTM and LSTM-p (due to the addition of the centroids), your observation that the state-regularized RNNs should be more constrained is absolutely correct. The training error on the truncated sequences of length 10 is indeed lower for the vanilla LSTMs.
>
>
> Comment (4): “MNIST experiments: please include results for SR-LSTM”
>
> We have included the results for the SR-LSTM in the revised version of the paper.
>
> We have also corrected the typo you mentioned in “Minor comments”
>
> We appreciate your insightful review which, as you can tell, has allowed us to improve the paper. Please let us know if there is anything else we can do.
>
>
> [1] Zaremba and Sutskever, Learning to Execute, 2015
> [2] Gail Weiss, Yoav Goldberg, and Eran Yahav. Extracting automata from recurrent neural networks using queries and counterexamples. 2018.

---

### Author Response · Authors · 2018-11-21
**The end of the discussion period is approaching**

Dear reviewers,

The period during which we can address your comments and make changes to the submission is coming to an end. We wanted to make sure that we have addressed your main concerns in a sufficient manner. Please let us know if you have additional suggestions for improvements. We'll be happy to incorporate those into the paper.  Thank you again for your valuable reviews and for considering our responses and revisions.

---

### Meta-Review · Area_Chair1 · 2018-12-14
**RBFN is back; a bit more work necessary**

**Confidence:** 4
**Recommendation:** Reject

**Metareview:**

the authors propose to incorporate an additional layer between the consecutive steps in LSTM by introducing a radial basis function layer (with dot product kernel and softmax) followed by a linear layer to make LSTM similar to or better at (by being more explicit) capturing DFA-like transition. the motivation is relatively straightforward, but it does not really resolve the issue of whether existing formulations of RNN's cannot capture such transition. since this was not shown theoretically nor intuitively, it is important for empirical evaluations to be thorough and clearly show that the proposed approach does indeed outperform the vanilla LSTM (with peepholes) when the capacity (e.g., the number of parameters) matches. unfortunately it has been the consensus among the reviewers that more thorough comparison on more conventional benchmarks are needed to convince them of the merit of the proposed approach.